# Study from microcosms and mesocosms reveals *Escherichia coli* removal in high rate algae ponds during domestic wastewater treatment is primarily caused by dark decay

Paul Chambonniere¤*, John E. Bronlund, Benoit Guieysse

Department of Chemical and Bioprocess Engineering, School of Food and Advanced Technology, Massey University, Palmerston North, New Zealand

¤ Current address: MicroAlgae Processes Platform CEA, CEA Tech Region Sud–Provence Côte d'Azur, Saint-Paul-lez-Durance, France
* paul.chambonniere@cea.fr

**Data Availability Statement:** All data, code, and matlab data sets files are available from the

## Abstract

While high rate algal ponds (HRAPs) can provide efficient pathogen removal from wastewater, the mechanisms involved remain unclear. To address this knowledge gap, the mechanisms potentially causing *Escherichia coli* (*E. coli*) removal during microalgae-based wastewater treatment were successively assessed using laboratory microcosms designed to isolate known mechanisms, and bench scale assays performed in real HRAP broth. During laboratory assays, *E. coli* decay was only significantly increased by alkaline pH (above temperature-dependent thresholds) due to pH induced toxicity, and direct sunlight exposure via UV-B damage and/or endogenous photo-oxidation. Bench assays confirmed alkaline pH toxicity caused significant decay but sunlight-mediated decay was not significant, likely due to light attenuation in the HRAP broth. Bench assays also evidenced the existence of uncharacterized 'dark' decay mechanism(s) not observed in laboratory microcosms. To numerically evaluate the contribution of each mechanism and the uncertainty associated, *E. coli* decay was modelled assuming dark decay, alkaline pH induced toxicity, and direct sunlight-mediated decay were independent mechanisms. The simulations confirmed *E. coli* decay was mainly caused by dark decay during bench assays (48.2–89.5% estimated contribution to overall decay at the 95% confidence level), followed by alkaline-pH induced toxicity (8.3–46.5%), and sunlight-mediated decay (0.0–21.9%).

## 1. Introduction

High rate algal ponds (HRAPs) are shallow raceway ponds that support efficient removal of organic carbon and nutrients during wastewater treatment. Few studies have shown HRAPs also support pathogen removal [1–3] at levels comparable to the efficiency achieved in maturation ponds, a well-established technology for wastewater disinfection [4–6]. Recent studies have confirmed this capacity at full-scale [7–9].

figshare database (doi: https://doi.org/10.6084/m9.figshare.c.5770658.v1).

**Funding:** The authors received no specific funding for this work.

**Competing interests:** The authors have declared no competing interests exist.

Various mechanisms drive pathogen removal in maturation ponds. Dark mechanisms include heat inactivation, sedimentation, predation, starvation/competition with other micro-organisms, and the toxicity caused by dissolved oxygen (DO), pH, and algal toxins [10,11]. Davies-Colley et al. (1999) [12] proposed that three sunlight-mediated mechanisms are the main drivers of pathogen removal in these systems, namely 1) direct absorption of solar UV-B by DNA resulting in DNA damage; 2) photo-oxidation of key cellular biomolecules by reactive oxygen species (ROS) generated by endogenous photosensitizers reacting with UV light; and 3) photo-oxidation of key cellular biomolecules by ROS generated by exogenous photosensitizers.

While the relative occurrence and significance of sunlight-mediated and dark removal mechanisms are currently unknown in HRAPs, sunlight-mediated mechanisms are expected to drive the bulk of pathogen removal based on studies in pilot scale systems [2,13]. However, Bahlaoui et al. (1998) [2] compared two HRAPs operated in parallel and found solar radiation was only significant in the HRAP operated at a constant hydraulic retention time. At full scale, Young et al. (2016) [8] found no relationship between F-RNA bacteriophage inactivation and solar radiation or water temperature, two parameters associated with this indicator inactivation in maturation ponds [14]. These authors also found no correlation between *Escherichia coli* (*E. coli*) removal and any of the parameters monitored. These results disagree with findings from maturation ponds, thus suggesting prevalent pathogen removal mechanisms may differ between HRAP and maturation ponds. Critically, the rapid and large temporal variability of key parameters experienced during pilot and full scale operations (e.g. influent flow and composition, temperature, incident sunlight intensity) impedes the determination of the mechanisms potentially involved, especially considering the different time scales involved.

These observations motivated the present study focusing on the removal of *E. coli*, the most commonly used indicator of wastewater disinfection [15], in HRAPs. To combine a deterministic laboratory scale approach with the representativeness of *in situ* investigations, this study was performed in three steps:

1. Laboratory assays (approx. 0.1 L) were first used to evaluate the potential magnitude of individual removal mechanisms under conditions representative of HRAP;

2. Bench assays (approx. 5 L) in HRAP liquid cultures including micro-organisms (henceforth referred to as HRAP broth for simplicity) were used to verify the relevance of laboratory observations under experimental conditions representative of real HRAP systems, while still enabling some control of broth conditions.

3. Due to large experimental uncertainty associated with microbial monitoring, the relative contributions and associated confidence intervals of the mechanisms identified in steps 1) and 2) were computed using a model based approach.

## 2. Materials and methods

### 2.1.Laboratory assays

Experiments investigating 'dark mechanisms' were conducted indoor under darkness in 150 mL E-flasks covered with foil, and experiments investigating 'light-mediated mechanisms' were conducted outdoor in opened 100 mL beaker. Sunlight experiments were performed on cloudless days to minimize the impact of sunlight variability caused by clouds. All E-flasks and beakers were autoclaved to minimize contamination interference, filled with 50 mL of auto-claved medium (unless otherwise notified) and aseptically inoculated with a wildtype *E. coli* strain to achieve an initial cell count between $2.0 \cdot 10^7$ and $2.0 \cdot 10^8$ CFU·mL$^{-1}$. This strain was

**Table 1. Summary of the conditions tested during laboratory assays.**

| Mechanism targeted | Experimental matrix | Chemical addition (Level) | Control of temperature (Range) | Indoor/ Outdoor | Duration (Range) |
|---|---|---|---|---|---|
| **Starvation and heat inactivation** | Reverse osmosis water | None | Yes (5–35°C) | Indoor | 2 hr—7 d[1] |
| **Toxicity of algal metabolites** | Filtrated HRAP broth | None | No | Indoor | 1 d -7d |
| **Wastewater toxicity** | Filtrated wastewater Centrifuged wastewater | None | No | Indoor | 2 d - 7d |
| **Ammonia toxicity** | Reverse osmosis water | pH buffer (pH 7–10) NH$_4$Cl (0.5–50 mg·L$^{-1}$) | Yes (10–35°C) | Indoor | 2 hr (35°C)– 7 hr (10°C) |
| **Alkaline induced pH-toxicity** | Reverse osmosis water | pH buffer (pH 7–11) | Yes (5–35°C) | Indoor | 0.2 hr (35°C, pH 11)– 2 d (5°C) |
| **Direct sunlight damage** | Reverse osmosis water | None pH buffer (pH 7 & 10) | No | Outdoor | 20 min—4 hr |
| **Exogenous photo-oxidation** | Filtrated wastewater Filtrated HRAP broth | None pH buffer (pH 7 & 10) | No | Outdoor | 20 min—2 hr |
| **Predation** | Not investigated in the present study | | | | |

[1] The duration varied since these conditions were used as controls for other experiments.

isolated from a pilot HRAP treating primary wastewater at Palmerston North wastewater treatment plant, New Zealand (Chambonniere et al. 2020 [16], see S1 Appendix). Following inoculation, the cultures were continuously agitated using a KS 260 control orbital shaker (IKA, Germany) at 200 rpm and incubated under specific conditions. Based on relevant literature [10], "natural decay" (starvation and heat inactivation), algal metabolite toxicity, wastewater toxicity, ammonia toxicity, alkaline-pH induced toxicity, sunlight direct damage (combining direct DNA damage and endogenous photo-oxidation), and exogenous photo-oxidation were investigated as potential *E. coli* decay mechanisms. The specific conditions used for each investigation are given in Table 1 with further details given in S2 Appendix. Predation was not investigated for practical reasons.

**Laboratory analysis.** Microbial counting was conducted using the pour-plate method [17]. The measurement uncertainty associated with log-transformed *E. coli* cell counts was estimated to be 4% (relative standard error, S3 Appendix). Temperature and pH were recorded using an Orion Star A326 multiprobe meter (Thermofisher, USA). Sunlight intensity data (10 minutes interval) was obtained from the National Institute of Water and Atmospheric Research Ltd database (Palmerston North, location agent number 21963).

## 2.2.Bench assays

Bench assays were developed to verify in real HRAP broth the significance of mechanisms demonstrated to impact *E. coli* survival during laboratory assays. Since pH toxicity and sunlight mediated mechanisms were the only mechanisms causing significant *E. coli* removal during laboratory assays (see Results and Discussion section), the bench assays were designed to focus on the parameters known to impact these mechanisms i.e. sunlight exposition, pH, and DO concentration (temperature could not be controlled). *E. coli* survival was therefore monitored under various combinations of 1) high or low pH, 2) high or low DO concentration, and 3) under sunlight radiation or in the dark.

For this purpose, two cylindrical reactors (25 cm deep, 14 and 16 cm diameters, 3.8 and 5.0 L working volumes) were set-up on the laboratory rooftop. The sides of the reactors were covered with opaque tape up to the 25 cm mark so that sunlight could reach the algae only from the liquid surface. The reactors were mixed using a vertical propeller (RW20 Janke & Kunkel, IKA, Germany) enabling water homogenization within 20 seconds based on visual colorimetric tests. The reactors were also equipped with an air bubbler (flat spiral coil) to control DO concentration.

Bench assays were conducted in broth collected from a pilot scale HRAP treating primary domestic wastewater located in Palmerston North, New Zealand having median broth total organic carbon, dissolved organic carbon, nitrate, ammonia, *E. coli* cell count levels of 94.2 mg·L$^{-1}$, 19.8 mg·L$^{-1}$, 51.5 mg·L$^{-1}$, 0.52 mg N·L$^{-1}$, and $1.48 \cdot 10^5$ MPN·100 mL$^{-1}$, respectively (set-up and operation fully described in [16]). Prior monitoring showed *Scenedesmus spp.* was the dominant species in the HRAP (unpublished data) but this was not verified during bench assays.

Various conditions of DO concentration ($> 10$ mg·L$^{-1}$ or $< 2$ mg·L$^{-1}$), pH ($> 9.4$ or $< 8$) were tested under natural sunlight exposure or in darkness. When DO concentration and pH were not left to increase naturally due to photosynthetic activity, DO concentration was reduced below 2 mg·L$^{-1}$ by bubbling N$_2$ gas through the air bubbler. In this case, the propeller was removed as gas bubbling also enabled reactor mixing. When needed, pH was kept neutral ($< 8$) by adding 0.1 M HCl.

**Assay preparation, sampling, and analysis.** On the day of each experiment (four days in total, all carried out in November 2017), HRAP broth was collected prior to 11 A.M. and immediately transported to the laboratory. Within 1 hour of collection, the two bench scale reactors were filled up with the HRAP broth to the 25 cm depth mark and inoculated with 2 mL of wild type *E. coli* suspension (S1 Appendix) to obtain a consistent initial *E. coli* cell count representative of levels typically reported in HRAP [16]. This inoculation time was considered as the start of the first bench assay, during which the reactors were exposed to sunlight for 2 hr (first light assay). The reactors were re-inoculated with 2 mL of wildtype *E. coli* suspension and exposed again to sunlight for 2 hr (second light assay). Following these two light assays, the reactors were re-inoculated with 2 mL of wildtype *E. coli* suspension, covered with cardboard, and incubated in darkness for another 2 hr (dark assay). An aliquot ($< 3$ mL) was collected from each reactor immediately following the addition of *E. coli* stock culture (time zero). The reactors were sampled every 30 minutes and *E. coli* cell count was quantified immediately following sample collection using the Quantitray® Colilert®-18 in accordance with the manufacturer instructions (IDEXX Laboratories, USA). Temperature, pH, and DO concentration were continuously logged at one minute intervals using an Orion Star A326 multiprobe meter (Thermofisher). Light attenuation coefficients were calculated from the measurement of HRAP broth transmittance for the wavelength 683 nm (PG Instrument Ltd UV/VIS Spectrophotmeter, T60) performed on each day bench assays were performed.

### 2.3. *E. coli* decay expression

In well-mixed batch reactors as operated during laboratory and bench scale experiments, the experimental rate of *E. coli* decay ($k$, d$^{-1}$) was computed from experimental data assuming first order kinetics [13,18,19] as:

$$\frac{dC}{dt} = -k(t) \cdot C(t) \qquad (1)$$

Where $C(t)$ represents *E. coli* cell count in the vessel (see S4 Appendix for further discussion). To quantify the removal efficiency of *E. coli* decay over a certain duration, the *E. coli* $\log_{10}$ removal was computed as $\log_{10}\left(\frac{C_0}{C_{end}}\right)$, where $C_0$ and $C_{end}$ are the initial and final *E. coli* cell count for the duration considered [20].

Determination and uncertainty during laboratory assays of thus determined *E. coli* decay rate are further detailed in S4 Appendix.

### 2.4. *E. coli* decay modelling

*E. coli* decay was mathematically modelled to quantify the relative contribution of mechanisms experimentally identified to cause significant *E. coli* removal. The model was used to estimate the confidence intervals associated with mechanism relative contributions.

*E. coli* decay was first modelled and parameterized based on results from laboratory assays. Changes in *E. coli* cell count during bench assays were then computed by calculating *E. coli* decay rate for every minute of the experiments (assuming the sunlight intensity was constant over 10-minute periods) using the Euler method [21] with a 1-minute time step from the first measured cell count, and assuming *E. coli* decay follows pseudo-first order kinetics. Algal broths were assumed to be well-mixed and shading from the mixer was neglected. Kinetic parameters of *E. coli* decay rate were corrected by fitting *E. coli* cell counts calculated for bench assays to the experimental dataset (kinetic parameters of *E. coli* decay rate are referred to as "fitted parameters" in the following). Model development, calibration, and fitting are described in the Results and Discussion section.

The uncertainty associated with the fitted parameters was assessed using Monte Carlo simulations [22]. For this purpose, experimental inputs associated with significant uncertainty were randomly varied within their range of uncertainty assuming normal or log-normal distribution (Table S5-1 in S5 Appendix) and new values for fitted parameters were each time computed. This operation was repeated 1,000 times (S5 Appendix).

The relative contribution of each mechanism to the overall *E. coli* decay and the confidence intervals associated with these relative contributions were assessed again using Monte Carlo simulations (S6 Appendix). Briefly, experimental measurements and fitted model parameters were randomly varied within their 95% confidence intervals and the relative contributions of the mechanisms studied (i.e. ratio between the number of cells decayed through a given mechanism over the total number of cells decayed during bench assays) were computed each time. This calculation was repeated 2,000 times.

### 2.5. Numerical and statistical analysis

All numerical and statistical analyses were performed using Matlab® R2019a (Mathworks Inc., Natick, MA, USA). Matlab® code files used to generate the results of this study can be found in the repository [23].

## 3. Results and discussion

### 3.1. Laboratory assays

**3.1.1. Laboratory assays in darkness.** *Starvation and heat inactivatio*. No significant reduction in *E. coli* viable cell counts (henceforth referred to as "decay" for simplicity) was recorded in darkness, in the absence of known harmful conditions, and at temperature up to 35°C (S7 Appendix). *E. coli* 'natural death' was therefore insignificant under the conditions tested, which confirms findings from Cook and Bolster (2007) [24] who reported a high *E. coli* survival in natural water (decay rate of 0.04 d$^{-1}$ over 400 days). *E. coli* optimally grows at

37.5˚C, the temperature of the human gut, and tolerates temperatures up to 48˚C [25]. As temperatures above 35˚C have hitherto not been reported during HRAP operations, starvation and heat inactivation are unlikely to be relevant to *E. coli* removal in full-scale HRAPs. While *E. coli* re-growth has been reported in some systems, particularly in warm tropical ponds [26], no significant increase of *E. coli* cell count was noticed during this study even at the warmest temperature tested (Fig S7-2 in S7 Appendix). Caution is therefore mandated before ruling out the possibility of regrowth, for *E. coli* or other indicators, depending on the conditions experienced.

*Algal-metabolite and wastewater toxicity*. Despite previous evidence that several microalgae can excrete a wide array of antibacterial compounds harmful to *E. coli* [27], *E. coli* decay was insignificant in flasks filled with algal filtrates. Algae-based toxicity was therefore unlikely significant under the conditions tested. This finding cannot be broadly extrapolated in view of the geographical and temporal variability in algal diversity found in HRAPs [28] and the diverse bactericidal compounds each species may be secreting [27,29]. *E. coli* decay was also insignificant in flasks supplied with filtered wastewater. Wastewater toxicity was therefore excluded as significant decay mechanism under the conditions tested.

*Ammonia toxicity*. No significant decay was recorded when $NH_4^+$ was supplied, even at pH 10 and concentration up to 50 mg·L$^{-1}$ (S8 Appendix). Considering the low $NH_4^+/NH_3$ concentration typically found in HRAP broths (generally $\leq$ 2 mg N-$NH_4^+$·L$^{-1}$ [30]), $NH_3$ toxicity is not expected to cause significant *E. coli* decay in HRAPs treating domestic wastewater. This is important as $NH_3$ has been suggested as a potential disinfectant in algal ponds [31,32] but direct evidence has to date been lacking.

*Alkaline-pH induced toxicity*. *E. coli* decay increased with pH, and this increase was temperature-dependent (S9 Appendix). *E. coli* pseudo-first order decay rate thus reached 67.5 ± 19.2 d$^{-1}$ at pH 10 and 30˚C (N = 8), which is significantly higher than the typical decay rates < 6 d$^{-1}$ at 20˚C in facultative and maturation ponds [32]. This discrepancy may be due to the fact that maturation ponds are characterized by lower pH conditions than HRAPs [33]. Because pH > 10 and temperatures > 25˚C can be experienced in HRAPs [34,35], alkaline-pH induced toxicity may indeed be significant in these systems. Such heightened decay rates agrees with findings from Parhad and Rao (1974) [36] who reported enhanced *E. coli* decay at pH > 9.4 in algal systems.

The exponential impact of pH on *E. coli* decay at constant temperature was described in a pseudo first order decay rate as:

$$\frac{dC}{dt} = -\left(a(T) \cdot 10^{pH-14}\right) \cdot C(t) \tag{2}$$

where $C(t)$ is *E. coli* cell count in a laboratory assay reactor (CFU·mL$^{-1}$), T is the broth temperature $T$ (˚C), $pH$ is the broth pH, and $a(T)$ is a temperature-dependent fitting parameter. The values of $a(T)$ were obtained as the slopes of the linear regressions of the decay rates against $10^{pH-14}$ at each temperature tested. The values of $a(T)$ were found to linearly increase with temperature when log-transformed ($R^2$ = 0.972, N = 6, Fig S9-3 in S9 Appendix), meaning the influence of temperature on *E. coli* decay rate at a given pH (i.e. $a(T)$) could be described by an Arrhenius equation [37] leading to the following final equation for pH induced toxicity to *E. coli*:

$$\frac{dC}{dt} = -(k_{20}^{pH} \cdot \theta^{pH\,T-20} \cdot 10^{pH-14}) \cdot C(t) \tag{3}$$

where $\theta^{pH}$ is the temperature-compensation coefficient for alkaline-pH induced toxicity and $k_{20}^{pH}$ is *E. coli* decay rate at 20˚C and pH 14. This temperature-dependent relationship can be explained by the hypothesis formulated by Mendonca et al. (1994) [38] that the effect of alkaline pH to *E. coli* is due to the solubilisation of membrane proteins or the saponification membrane lipids, two

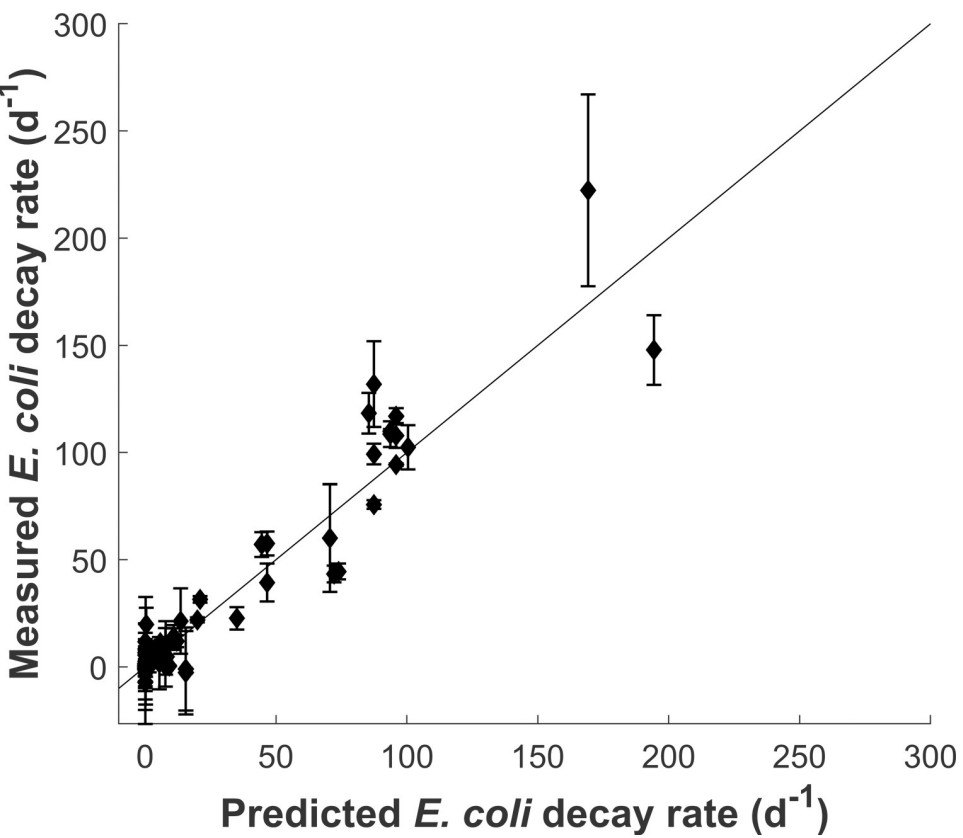

**Fig 1. Measured versus fitted *E. coli* decay rates for all tested pH between 6.5 and 10.5 and temperatures between 5 and 35°C in laboratory assays.** The 45° line shows equality between measured and predicted data. Error bars show the standard error of linear regression performed.

reactions which temperature-dependence have been modelled using the Arrhenius equation [39,40]. The numerical values of $\theta^{pH}$ (1.14) and $k_{20}^{pH}$ (1.49·10$^5$ d$^{-1}$) were determined from the linear regression of ln($a$) with $T$. As can be seen in Fig 1, Eq 3 satisfyingly fitted experimental data (coefficient of determination between measured and predicted data = 0.914, N = 81). Because of the very fast decay occurring at pH above 10.5, only a few experiments performed under these conditions yielded measurable results, and the values obtained suffered from high uncertainty, particularly at higher temperature (e.g. only two values of decay rate could be calculated at pH > 10.5 at 35°C: 1212 and 1713 d$^{-1}$; not shown in Fig 1). Consequently, Eq 3 was calibrated excluding such data and may not accurately predict *E. coli* decay at pH > 10.5.

**3.1.2. Laboratory assays under natural sunlight.**   *Sunlight irradiation.* In open beakers exposed to sunlight at neutral pH, *E. coli* log$_{10}$ removal was linearly correlated with sunlight dose without intercept (0.314 ± 0.0212 m$^2$·MJ$^{-1}$, R$^2$ = 0.920, N = 20, p = 6.84·10$^{-12}$, Fig 2). This correlation yields a relationship between *E. coli* cell count rate (***C(t)***, CFU·mL$^{-1}$) and incident sunlight intensity (***Hs***, W·m$^{-2}$) that can be expressed as:

$$\frac{dC}{dt} = -(\alpha \cdot Hs(t)) \cdot C(t) \qquad (4)$$

Where $\alpha$ is the sunlight specific decay rate due to direct sunlight damage (6.24·10$^{-2}$ ± 4.2·10$^{-3}$ m$^2$·W$^{-1}$·d$^{-1}$). The sunlight specific decay rate due to direct sunlight damage predicted

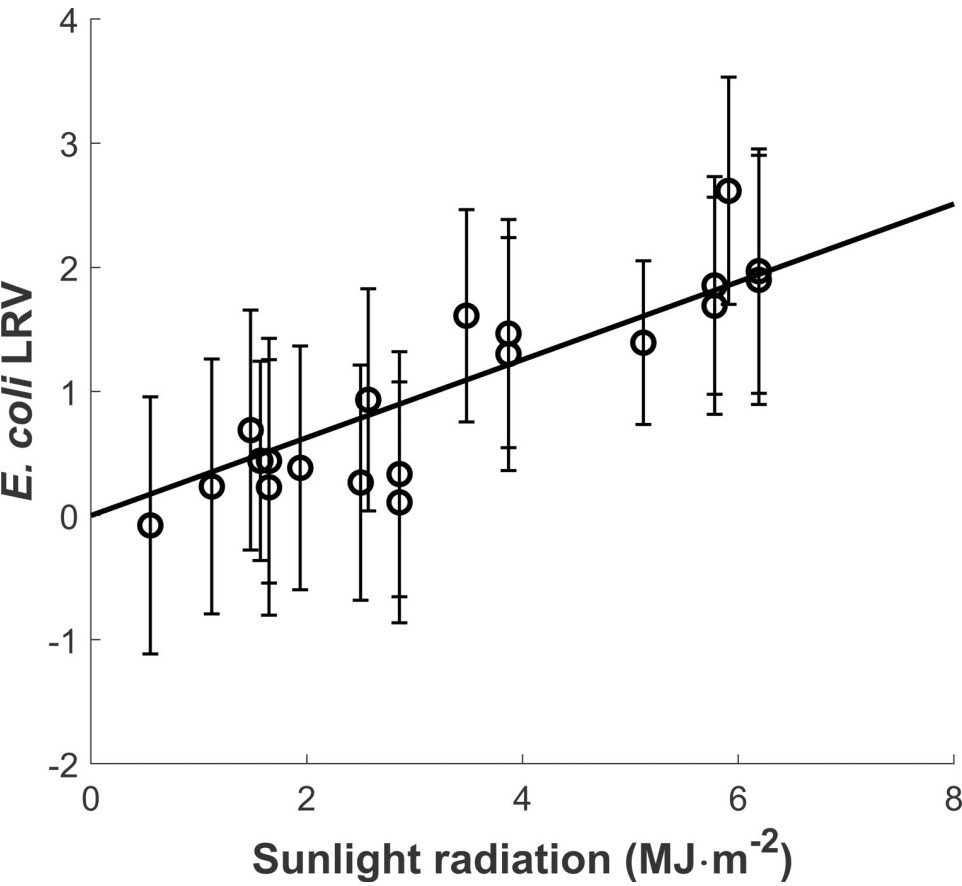

**Fig 2. Effect of sunlight dose on *E. coli* log$_{10}$ removal at neutral pH during laboratory assays.** The straight line represents the linear regression with null intercept peformed over the dataset (slope converted to the sunlight specific first order decay rate of 0.0624 m$^2$·W$^{-1}$·d$^{-1}$ by multiplying by the factor $24 \cdot 3600$ $[s/d]/(10^6[MJ/J]) \cdot ln(10)$. An outlier confirmed by Grubb's test on the residuals from linear regression was removed from the analysis. Error bars show measurement standard error.

in the present study is challenging to compare to existing studies using different experimental conditions (e.g. strain as shown in S10 Appendix, experimental broth matrix, radiation wavelength) and/or because the inputs needed to calculate the sunlight dose and the log$_{10}$ removal (or alternatively, sunlight intensity and first order decay rates) are not always provided in the literature. Nevertheless, Maraccini et al. (2016a) [41] determined a sunlight specific decay rate of 7.01 m$^2$·W$^{-1}$·d$^{-1}$ for *E. coli* exposed to UV-B in natural waters. Assuming UV-B accounts for 5% of the total UV radiation reaching the Earth surface and UV radiation accounts for approx. 5% of the total solar spectrum [42], this translates to a sunlight specific decay rate of $1.75 \cdot 10^{-2}$ m$^2$·W$^{-1}$·d$^{-1}$ which is in the order of magnitude predicted in the present study. Curtis et al. (1992b) [43] reported a sunlight specific decay rate of $1.07 \cdot 10^{-2}$ m$^2$·W$^{-1}$·d$^{-1}$ (assuming pH and DO concentration are constant) for faecal coliforms incubated in filtrated algae pond water. This value is again in the order of magnitude of the rate herein reported for *E. coli* in reverse osmosis water. A direct comparison of the rate values is however difficult given the different microbial indicator used.

*Influence of pH.* In agreement with Davies-Colley et al. (1999) [12], *E. coli* decay under sunlight was significantly enhanced at high pH (S11 Appendix). It was however not possible to determine if the high pH contributed to additional *E. coli* removal, or if it synergistically

enhanced sunlight-mediated removal. *E. coli* decay rates under sunlight at alkaline pH observed were higher than hitherto reported [31,44] which may be explained by the higher pH and temperature used in the present work compared to these past studies.

*Impact of photosensitizers.* Decay rates recorded in reverse osmosis water did not statistically differ from decay rates recorded in wastewater or HRAP filtrates (one-tailed paired two-sample t-test at the 5% significance level, N = 9, p = 0.584, S12 Appendix). Overall, these tests suggest *E. coli* decay under sunlight is not significantly improved by the probable presence of photosensitizers under the conditions tested. Therefore, and despite the potential existence of a shoulder lag-period before exogenous photo-oxidation would become significant [18,45,46], this mechanism was unlikely significant for *E. coli* in HRAP (S12 Appendix). It is possible that instead (or in spite) of leading to the creation of ROS species, dissolved substances in the HRAP filtrates were absorbing sunlight and thereby reduced sunlight-mediated damage to *E. coli* cells (the sunlight absorption spectra of the filtrates were not recorded during the experiments but an example of absorption spectrum for wastewater and HRAP filtrates can be found in S12 Appendix). Regardless, our finding agrees with the conclusion of Maraccini et al. (2016b) [46] who found gram-negative bacteria such as *Salmonella enterica*, *E. coli* K-12, and *E. coli* O157:H7 had a high resistance to exogenous photo-oxidation in presence of naturally occurring organic photosensitizers. The specific impact of dissolved oxygen (DO) concentration on sunlight-mediated removal could not be practically investigated in laboratory assays due to technical difficulties in reaching HRAP supersaturated DO levels (up to 300% of the atmospheric saturation values [34]) under aseptic conditions. The addition of known photosensitizers was not tested as this manipulation would at best only verify that these known chemicals generate radicals enhancing *E. coli* removal, without evidencing that the mechanism is indeed taking place in the absence of the added chemicals.

### 3.2. Bench assays

Four bench scale experiments were conducted in HRAP broth. Each experiment was duplicated (Reactors A and B) and included a series of three sub-experiments conducted under various conditions of light, pH and DO concentration (Table 2). To confirm the potential significance of alkaline pH induced toxicity and photo-oxidation, *E. coli* $\log_{10}$ removal after each sub-experimental phase were analysed against the total sunlight dose received (MJ·m$^{-2}$), and the averaged pH, DO, and temperature recorded. A complementary analysis based on *E. coli* decay rates (d$^{-1}$) is provided in S13 Appendix.

**E. coli decay in the dark.** At neutral pH, *E. coli* $\log_{10}$ removal values were similar in darkness and under sunlight during Experiment 2, but higher in darkness during Experiments 1 and 3. This similarity suggests that mechanisms causing decay under darkness were also responsible for *E. coli* removal under sunlight. During Experiment 4, *E. coli* cells introduced into the HRAP broth were inactivated within minutes in darkness at neutral pH and high DO concentration. As these very high decay rates were recorded in the absence of any of the harmful conditions identified during laboratory scale experiments, these results further evidence that previously undetected dark removal mechanisms drove *E. coli* decay in real algal broth.

**Impact of sunlight exposure.** During laboratory assays in clear medium, only sunlight was found to influence *E. coli* survival at neutral pH. Yet, in bench assay conducted in algal broth, similar *E. coli* $\log_{10}$ removal values were observed in the dark and under sunlight (e.g. Reactor B, Experiment 2). Assuming the removal efficiency of dark mechanisms is not negatively impacted by light exposure, the similar removals reported during dark and illuminated bench assays suggests that sunlight mediated decay mechanisms had little impact on *E. coli* decay in the full algal broth. The analysis of *E. coli* decay rate also confirmed this conclusion (S13 Appendix).

**Table 2. Results from bench experiments by experiment.**

| Assay | | Temperature (˚C) | | DO (mg·L$^{-1}$) | | pH | | Light Dose (MJ·m$^{-2}$) | Time Interval | *E. coli* log$_{10}$ removal [1] | |
|---|---|---|---|---|---|---|---|---|---|---|---|
| | | Reactor | | Reactor | | Reactor | | | | Reactor | |
| Experiment | Test | A | B | A | B | A | B | | | A | B |
| Experiment 1 | 1 | 25.5 | 25.4 | 17 | 1.5 | 9.6 | 9.5 | 4.95 | 1:50 | 1.48 | 1.24 |
| | | | | | | | | | | [1.15–1.80] | [0.95–1.57] |
| | 2 | 30 | 27.7 | 23 | 1.3 | 10.4 | 10.5 | 3.06 | 1:30 | 2.77 | 1.34 |
| | | | | | | | | | | [2.40–3.18] | [1.19–1.47] |
| | 3 | 31 | 28.2 | 20 | 0.40 | 10.7 | 10.8 | 0 | 1:05 | 3.01 | 1.97 |
| | | | | | | | | | | [2.24–2.98] | [1.29–2.03] |
| Experiment 2 | 1 | 24.3 | 22.9 | 22 | 1.2 | 7.3 | 7.5 | 5.64 | 1:55 | 1.09 | 1.32 |
| | | | | | | | | | | [0.77–1.40] | [0.95–1.70] |
| | 2 | 28.6 | 27.3 | 31 | 0.80 | 7.4 | 6.9 | 4.91 | 1:50 | 1.75 | 1.28 |
| | | | | | | | | | | [1.37–2.12] | [0.94–1.60] |
| | 3 | 31.1 | 29.6 | 24 | 0.50 | 6.3 | 7.6 | 0 | 1:40 | 1.61 | 1.23 |
| | | | | | | | | | | [1.26–2.01] | [0.85–1.60] |
| Experiment 3 | 1 | 27.7 | 26.9 | 24 | 15 | 10.4 | 7.3 | 6.95 | 2:00 | 2.47 | 1.61 |
| | | | | | | | | | | [2.07–2.88] | [1.26–1.97] |
| | 2 | 32 | 31.7 | 28 | 23 | 11.2 | 6.9 | 5.97 | 1:50 | 2.83 | 1.59 |
| | | | | | | | | | | [2.44–3.24] | [1.18–2.06] |
| | 3 | 34 | 33.8 | 20 | 24 | 10.9 | 7.0 | 0 | 0:45[2] | > 2.61[2] | 1.22 |
| | | | | | | | | | | | [0.85–1.65] |
| Experiment 4 | 1 | 31.3 | 30.3 | 24 | 21 | 10.3 | 7.2 | 7.15 | 2:00 | 3.66 | 1.24 |
| | | | | | | | | | | [2.94–4.63] | [0.93–1.55] |
| | 2 | 36.6 | 36.8 | 25 | 26 | 10.4 | 7.1 | 2.84 | 0:50[2] | > 4.05[2] | 0.59 |
| | | | | | | | | | | | [0.19–1.00] |
| | 3 | 39.6 | 38.3 | 18 | 25 | 10.4 | 6.3 | 0 | 0:00[3] | >>[3] | >>[3] |

[1] The values in bracket show the 95% confidence interval calculated based on the Quanti-Tray MPN table uncertainty.

[2] No live *E. coli* cells were found in the second sample withdrawn at the dilution tested. A minimal log removal was calculated based on the first cell count measured and the analytical detection limit of Quanti-Tray countings.

[3] No *E. coli* were found in the first sample (i.e. within 10 minutes following the cells suspension in the algal broth) so no removal efficiency could be measured (log removal > 4 within minutes can be inferred from these measurements).

**Impact of DO concentration.** While DO concentration had no apparent impact on *E. coli* decay at neutral pH (Experiment 2), high DO concentrations were associated with improved *E. coli* decay at high pH (Experiment 1). Critically, these differences (or lack of) were reported regardless of if the flasks were exposed to light or incubated in darkness, which disagrees with past literature showing DO concentration impacts *E. coli* via the formation of ROS (reactive oxygen species) under sunlight radiation but does not impact *E. coli* survival in the dark [31,44]. Light-mediated ROS formation should have therefore enhanced *E. coli* decay only under light exposure and this effect should have been see in both Experiments 1 and 2 under sunlight, as opposed to the present observations showing no impact of light, but an interplay between the impacts of pH and DO concentration (Experiment 1 versus 2). The positive impact of DO concentration at high pH (Experiment 1) was instead likely due to the higher temperatures experienced in Reactor A than Reactor B. Indeed, Reactor A was slightly smaller than Reactor B, meaning higher temperatures were typically experienced in Reactor A (high DO) than in Reactor B (low DO). Since alkaline pH-induced toxicity to *E. coli* is sensitive

to temperature (as evidenced during laboratory assays), temperature likely enhanced alkaline pH-induced toxicity more in Reactor A than in Reactor B during Experiment 1 conducted at high pH. This positive impact of temperature on alkaline pH-induced toxicity was logically not significant during Experiment 2 conducted at neutral pH. DO concentration is therefore concluded to have no significant impact on *E. coli* removal under the conditions studied.

**Impact of pH.** During Experiment 3, *E. coli* removal under sunlight was significantly faster at pH 10.4–10.8 than at neutral pH, and *E. coli* removal in darkness was significantly faster at pH 10.9 than at neutral pH. These results confirmed laboratory findings that alkaline pH induced toxicity is a significant contributor to *E. coli* decay. Statistical analysis of the decay rate dataset also showed with high confidence that pH had a positive impact on *E. coli* decay (S13 Appendix).

In comparison with results from laboratory assays, bench assays confirmed that *E. coli* decay was enhanced by elevated pH but evidenced that sunlight-mediated decay had limited impact in HRAP broth. The low impact of sunlight on *E. coli* during bench assays was likely caused by a high light attenuation in the HRAP broths (light attenuation coefficients in the range 55–70 m$^{-1}$ at 683 nm). This attenuation was itself caused by the presence of pigmented algae cells and light attenuation can be expected to be even stronger for the UV radiation known to be lethal to *E. coli* [13] than for visible wavelength such as 683 nm [47]. The elevated *E. coli* decay recorded during bench assays at neutral pH, both in the dark and under sunlight, was therefore likely caused by dark mechanisms. Since such dark mechanisms went undetected during laboratory assays, the use of real HRAP broth could have induced conditions causing microalgae to release antibiotics during bench assays. For example, the presence of competitive organisms, high DO levels, osmotic stress, and UV exposure (HRAP broth was exposed to sunlight 4 hours before being placed in darkness) have all been reported to generate stresses that can cause microalgae to secrete bactericidal compounds [48]. The possible impact of predation could not be practically assessed in our study but this mechanism has been reported to have a major effect on bacterial removal in waste stabilization ponds [10] and could have explained the unexpectedly high decay recorded in the dark during bench assays.

## 3.3. Model development from bench assays

There was no clear evidence that alkaline-pH induced toxicity and sunlight-mediated decay were interacting during laboratory assays (S11 Appendix). Consequently, the total *E. coli* decay rate from bench microcosms was first computed as the sum of the alkaline-pH induced toxicity (Eq 3) and sunlight-mediated decay (Eq 4 modified to account for light attenuation from suspended solids found in HRAPs, S14 Appendix), using their expressions as established and parameterized from the laboratory data. This model performed poorly against bench data ($R^2$ = -2.55, N = 54), which was not surprising because bench assays evidenced significant dark decay mechanisms and low impact of sunlight thereby evidencing the lack of representativeness of laboratory assays. Since bench assay evidenced dark decay was significant, a mechanisms that is likely temperature-dependent, the decay model was modified by adding an Arrhenius expression for dark decay rate as commonly done for bacteria removal in maturation ponds [37,49,50]. The laboratory-based model also overestimated *E. coli* decay at elevated pH, suggesting alkaline-pH induced toxicity was overestimated. Because microbial communities form biofilms that protect them against various adverse factors [51], including moderate pH [52–54], the presence of solids and other microbial species in the HRAP microcosms likely protected *E. coli* from alkaline-pH induced toxicity during bench assays. To account for this effect, the relevant pH-model parameters were recalibrated against bench data as described

below. Following these modifications, *E. coli* cell count in bench reactors (*C*) was computed as:

$$\frac{dC}{dt} = -\left( k_{20}^{dark} \cdot \theta^{dark\,T-20} + k_{20}^{pH} \cdot \theta^{pH\,T-20} \cdot 10^{pH-14} + \frac{\alpha.Hs(t)}{\sigma.d} \cdot \left(1 - e^{-\sigma.d}\right) \right) \cdot C \qquad (5)$$

where $k_{20}^{dark}$ is *E. coli* dark decay coefficient at 20˚C, $\theta^{dark}$ is the temperature compensation coefficient for dark decay, *d* is the water column depth (m), and $\sigma$ the light extinction coefficient of the algal broth (m$^{-1}$). Values (and associated uncertainties) of the experimentally measured parameters (*C*, *pH*, T, *Hs*, *d* and $\sigma$) are summarized in Table S5-1 in S5 Appendix.

The values of the fitted parameters $k_{20}^{dark}$, $\theta^{dark}$, $k_{20}^{pH}$, $\theta^{pH}$, and $\alpha$ were computed by fitting the model outputs against specific subsets of bench data focusing on specific removal mechanisms. The model was initialized by implementing parameter values derived from laboratory data (i.e. $k_{20}^{pH} = 1.49 \cdot 10^5$ d$^{-1}$, $\theta^{pH} = 1.14$, $k_{20}^{dark} = 0$ d$^{-1}$, $\theta^{dark} = 1$, $\alpha = 0.0624$ m$^2$·W$^{-1}$·d$^{-1}$) and a fitting algorithm was performed following these successive computation steps:

1. The values of $k_{20}^{dark}$ and $\theta^{dark}$ were computed by minimizing the sum of squared residuals on the neutral pH data subset when varying $k_{20}^{dark}$ and $\theta^{dark}$ (all other parameters being kept constant);

2. The values of $k_{20}^{pH}$ and $\theta^{pH}$ were computed by minimizing the sum of squared residuals on the elevated pH data subset when varying $k_{20}^{pH}$ and $\theta^{pH}$ (all other parameters being kept constant);

3. The value of $\alpha$ was computed by minimizing the sum of squared residuals over the entire data set when varying $\alpha$ (all other parameters being kept constant).

This algorithm was repeated until all fitted parameters did not vary relatively by more than 1%. The values thus obtained for fitted parameters are shown in Table 3 (labelled as 'best fit'). The average relative error over full bench data set corresponding to the best fit parameters was 6.0% ($R^2 = 0.812$, N = 54, Fig 3). As can be seen, Eq 5 henceforth calibrated could reproduce *E. coli* cell counts during bench experiments in HRAP broth with good accuracy. Further calibration using tests performed in HRAP broth collected at other times of the year (e.g. winter) and independent validation in a real HRAP are still needed before the model can be used to predict HRAP disinfection performance.

A Monte-Carlo analysis was carried out to determine the impact of uncertainties in the data used during model parameterization (Table S5-1 in S5 Appendix) on the computation of $k_{20}^{dark}$, $\theta^{dark}$, $k_{20}^{pH}$, $\theta^{pH}$, and $\alpha$ (i.e. the fitted parameters). Because the distributions of fitted model parameters were not necessarily normal (Fig S5-1 in S5 Appendix and S6-2 in S6 Appendix), median, mean, 5, and 95 percentiles of the data calculated are shown (Table 3). As can be seen, large uncertainty is associated with the parameterization of the model and an additional

**Table 3. Model parameters uncertainty estimated by Monte Carlo analysis.**

| Parameter | $k_{20}^{dark}$ (d$^{-1}$) | $\theta^{dark}$ (-) | $k_{20}^{pH}$ (d$^{-1}$) | $\theta^{pH}$ (-) | $\alpha$ (m$^2$·W$^{-1}$·d$^{-1}$) |
|---|---|---|---|---|---|
| Best fit | 47.4 | 1.00 | $2.86 \cdot 10^3$ | 1.45 | 0 |
| Median | 39.6 | 1.00 | $3.40 \cdot 10^3$ | 1.42 | 0 |
| Average | 37.6 | 1.02 | $7.72 \cdot 10^3$ | 1.39 | $1.13 \cdot 10^{-1}$ |
| 5 Percentile | 9.98 | 1.00 | $2.22 \cdot 10^3$ | 1.19 | 0 |
| 95 Percentile | 58.4 | 1.10 | $2.58 \cdot 10^4$ | 1.49 | $4.91 \cdot 10^{-1}$ |
| Laboratory assays | 0 | 1.00 | $1.49 \cdot 10^5$ | 1.14 | $6.24 \cdot 10^{-2}$ |

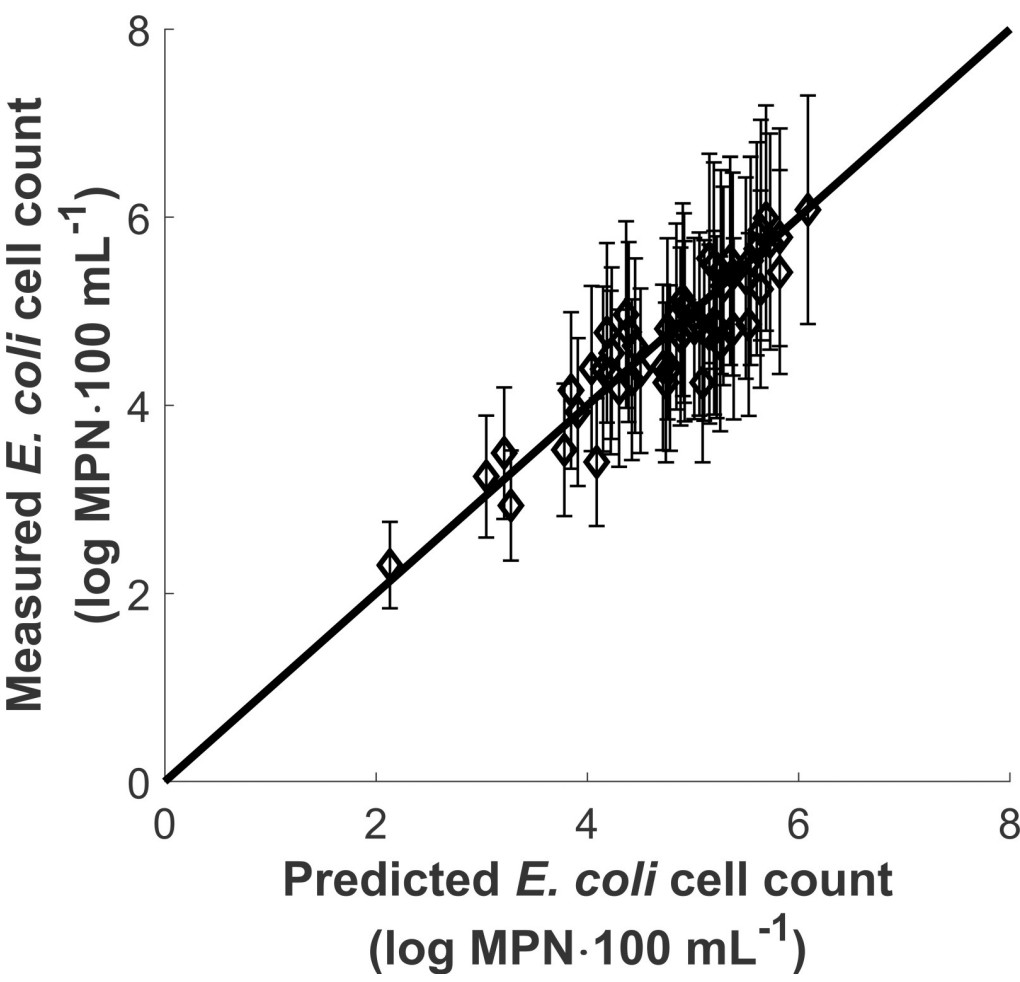

**Fig 3. Measured vs. computed *E. coli* cell counts during bench assays using 'best-fit' model parameters ($R^2$ = 0.812, N = 54).** Error bars show 20% uncertainty on measured *E. coli* cell counts.

sensitivity analysis (S15 Appendix) showed that this uncertainty was primarily caused by *E. coli* cell counts measurement uncertainty (inherent to Quantitray® Colilert®-18 method).

### 3.4.Mechanistic implications

The data shown in Table 3 suggest uncharacterized dark decay was not impacted by temperature ($\theta^{dark}$ = 1 for best fit) within the temperature range tested (19.9–38.1°C), though $\theta^{dark}$ uncertainty range includes values up to 1.10 underlining that high temperature dependence for this mechanism is possible. A broader range of temperature should be tested to refine this finding. Alkaline-pH induced toxicity was confirmed to be less effective during bench assays than during laboratory assays (significant decrease of $k_{20}^{pH}$), but was more sensitive to temperature (significant increase of $\theta^{pH}$). Critically, sunlight-mediated decay has no predicted impact on *E. coli* removal as the model best fit was obtained when the value of $\alpha$ was null.

The relative contribution (%) of each single *E. coli* decay mechanism to overall *E. coli* removal during bench assays was further assessed using Monte-Carlo analysis to account for the impact of uncertainties due to experimental error (Table S5-1 in S5 Appendix) and parameterization uncertainty (Table 3). Based on the 5–95 percentiles of the values thus calculated (Fig 4),

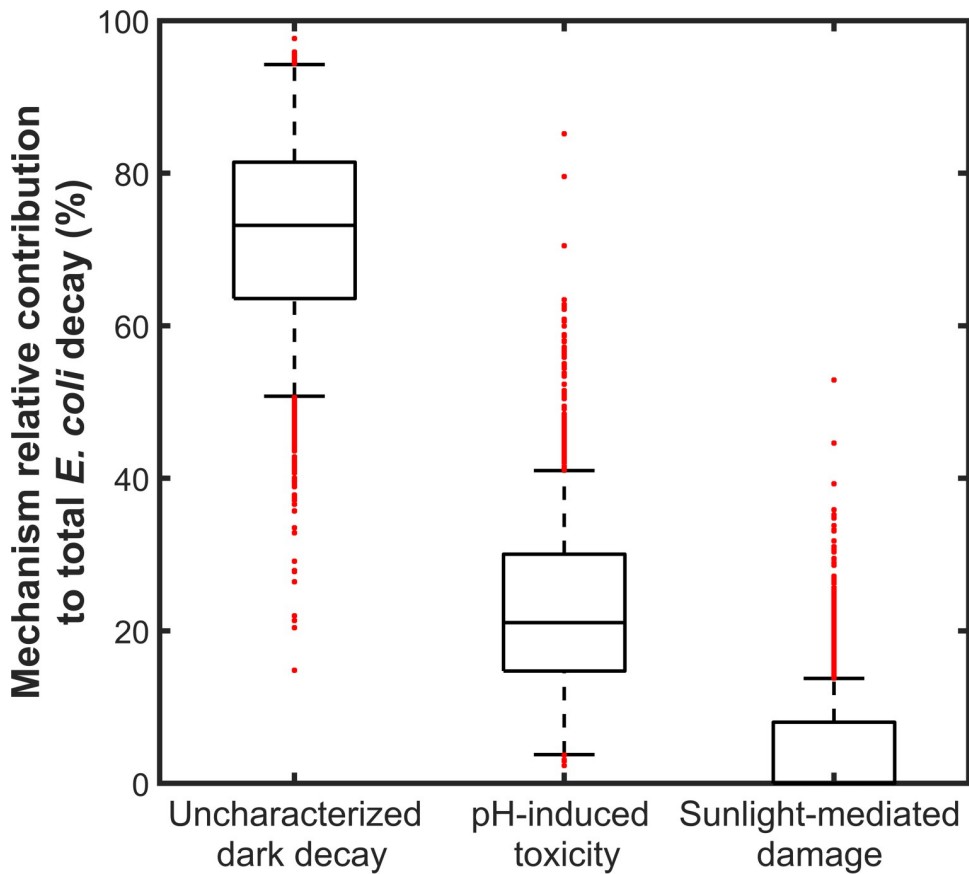

**Fig 4. Relative contributions of removal mechanisms contributing to *E. coli* decay during bench assays.** Boxplots represent the 5, 25, 50, 75, and 95 percentiles of the distributions, and red dots the outlying data (N = 1322).

uncharacterized dark decay was found to account for 48–89% of the overall *E. coli* decay during bench assays, against 8–46% for alkaline-pH induced toxicity, and 0–22% for sunlight-mediated removal. This analysis therefore showed with strong confidence that direct sunlight damage contributed little to *E. coli* removal during bench assays, while uncharacterized dark decay was likely the main contributor. This finding is critical because sunlight is commonly viewed as the main factor of pathogen removal during wastewater treatment in HRAPs [13,55].

Further research is critically needed to better understand the uncharacterized dark mechanisms involved, as this knowledge may provide the foundation for significantly improving HRAP design and operation for pathogen removal. The specific mechanisms causing *E. coli* removal under darkness were not identified during this study. Because predation (e.g. from protozoa or coliphages) has been reported as a main mechanism of bacterial decay in the dark [10], investigating this dark mechanism in the context of HRAP is of utmost interest, especially considering the predominance of dark decay herein reported. A significant contribution to overall dark decay from bactericidal compounds produced by microalgae cannot be ruled out although it was never observed in specific laboratory assays. It is also critical to consider the findings from this study may not apply to HRAPs operated under significantly different conditions (e.g. different climates, pH control through $CO_2$ addition, [56]) and/or hosting different microalgae ecology (e.g. influencing pH variations [57], potentially generating toxic metabolites [48]) or, especially, to different pathogens or indicators than *E. coli*. For instance, gram-positive indicators were found to better resist alkaline-pH induced toxicity likely due to

differences in the cell membrane structure [38], while gram positive bacteria were on the contrary reported to be more susceptible to exogenous photo-oxidation facilitated by naturally occurring organic photosensitizers [46]. Similar studies should therefore be carried out to investigate the removal of different indicators during HRAP based wastewater treatment.

## 4. Conclusions

The present study showed with high confidence that dark mechanisms caused most (48–89%) of *E. coli* decay in HRAP mesocosms while sunlight mediated mechanisms only caused a negligible to limited removal (0–22%). Alkaline-pH induced toxicity was the only other significant *E. coli* decay mechanism identified, causing 8–46% of total decay. The exact underlying mechanisms of *E. coli* decay in the dark were not identified, and further research to characterize dark decay in HRAP is critically needed as it could be the foundation for significant improvement of HRAP disinfection performance.

Finally, while some conditions reported to be harmful to *E. coli* correlated well with *E. coli* decay at laboratory scale, such correlations were less evident (high pH) or non-existent (sunlight intensity) at bench scale. This study therefore underlines that controlled experiment to identify conditions of microbial decay can lack the representativeness of real conditions. Future investigations of the mechanisms of microbial decay during wastewater treatment in HRAPs need to be carried out in broth and under conditions as comprehensively representative of HRAP culture as practical.

## Supporting information

**S1 Appendix. *E. coli* strain used in the study: Isolation, cultivation, and description.**
(PDF)

**S2 Appendix. Detailed laboratory assay protocols.**
(PDF)

**S3 Appendix. Uncertainty of cell count obtained from pour-plate method.**
(PDF)

**S4 Appendix. Calculation of first order decay rate and associated uncertainty during laboratory assays.**
(PDF)

**S5 Appendix. Description of Monte Carlo method for the determination of fitted model parameters uncertainty.**
(PDF)

**S6 Appendix. Uncertainty associated with the relative contribution of studied mechanisms to *E. coli* decay during bench assays.**
(PDF)

**S7 Appendix. *E. coli* starvation and heat inactivation during laboratory assays.**
(PDF)

**S8 Appendix. Ammonia toxicity to *E. coli* during laboratory assays.**
(PDF)

**S9 Appendix. Alkaline-pH induced toxicity to *E. coli* during laboratory assays.**
(PDF)

**S10 Appendix. Decay of laboratory versus wildtype *E. coli*.**
(PDF)

**S11 Appendix. Interaction between elevated pH and sunlight mediated *E. coli* decay.**
(PDF)

**S12 Appendix. Impact of photosensitizers on *E. coli* decay under sunlight during laboratory assays.**
(PDF)

**S13 Appendix. *E. coli* decay during bench assays: Relationship with environmental parameters.**
(PDF)

**S14 Appendix. *E. coli* decay due to sunlight direct damage in opaque broth: Model development.**
(PDF)

**S15 Appendix. Sensitivity analysis of *E. coli* decay rate modelling in HRAP broth.**
(PDF)

## Acknowledgments

The authors would like to acknowledge Zoe Foreman for her important contribution in the study of *E. coli* decay under alkaline pH conditions and according to ammonia concentration during laboratory assays. This particular work was performed as part of her Bachelor of Engineering thesis (Hons.) at Massey University.

The authors would also like to thank Professor Felipe Bravo Oviedo (Universidad de Valladolid) for providing hardware which facilitated the modelling work performed in this study, and Dr. Maxence Plouviez for his help to extract and sequence the DNA of the *Escherichia coli* strain used during this work.

## Author Contributions

**Conceptualization:** Paul Chambonniere, John E. Bronlund, Benoit Guieysse.

**Data curation:** Paul Chambonniere, John E. Bronlund.

**Formal analysis:** Paul Chambonniere, Benoit Guieysse.

**Funding acquisition:** Benoit Guieysse.

**Investigation:** Paul Chambonniere.

**Methodology:** Paul Chambonniere, John E. Bronlund, Benoit Guieysse.

**Project administration:** Benoit Guieysse.

**Software:** Paul Chambonniere.

**Supervision:** John E. Bronlund, Benoit Guieysse.

**Validation:** Benoit Guieysse.

**Visualization:** Paul Chambonniere, John E. Bronlund.

**Writing – original draft:** Paul Chambonniere.

**Writing – review & editing:** Paul Chambonniere, John E. Bronlund, Benoit Guieysse.

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
