## [Decision Letter · Decision Letter 0]

8 Feb 2022

PONE-D-21-40922Study from microcosms and mesocosms reveals Escherichia coli removal in high rate algae ponds during domestic wastewater treatment is primarily caused by dark decayPLOS ONE

Dear Dr. Paul,

Thank you for submitting your manuscript to PLOS ONE. After careful consideration, we feel that it has merit but does not fully meet PLOS ONE’s publication criteria as it currently stands. Therefore, we invite you to submit a revised version of the manuscript that addresses the points raised during the review process.

We look forward to receiving your revised manuscript.

Kind regards,

Dr. Harish

Academic Editor

PLOS ONE

Journal Requirements:

Reviewers' comments:

Reviewer's Responses to Questions

**Comments to the Author**

1. Is the manuscript technically sound, and do the data support the conclusions?

Reviewer #1: Yes

Reviewer #2: Yes

2. Has the statistical analysis been performed appropriately and rigorously? 

Reviewer #1: Yes

Reviewer #2: Yes

3. Have the authors made all data underlying the findings in their manuscript fully available?

Reviewer #1: Yes

Reviewer #2: Yes

4. Is the manuscript presented in an intelligible fashion and written in standard English?

Reviewer #1: Yes

Reviewer #2: Yes

5. Review Comments to the Author

Reviewer #1: Authors have presented a very interesting study on targeting a protocol for removal of pathogenic bacteria from wastewater used as a medium for growth of microalgae. I have few queries to which authors may reply:

1. Why only one bacteria was tested since in wastewater there are other pathogenic and non pathogenic bacteria present too.

2. If the study showed that E coli decay at alkaline pH and dark decay mechanism which although authors have shown statistically , however, to anyone studying the algae related cultivation using wastewater understanding the physiology of this study is very important. Authors should add up what is the biology behind the adaption of this behavior.

3. Further another important question is whether this study is useful for other algae also or only for Scenedesmus spp. Why many microalgae species which grow under alkaline conditions face bacterial infection, authors may throw some reasoning at this aspect so that the technology and protocol can be adapted by the readers for other algae too.

Reviewer #2: High-rate algae ponds (HRAPs) are wastewater treatment systems that enable combining cost-efficient secondary treatment at small scale with the production of a harvestable biomass for subsequent valorisation (e.g. biofuel). However, there is still limited data on pathogen removal during long-term HRAP operation with real effluents. This manuscript showed the potential significance of mechanisms driving pathogen removal in lab scale and bench scale in light of the specific environmental conditions occurring in HRAPs. However, Authors should improve this by following comments:

Starvation and heat inactivation:

You have good observation in this section but even at less temperature E coli including other bacteria can still survive and flourish in photobioreactors with monoculture of microalga. Please justify with more references.

Algal-metabolite section is still unclear please addon this with few references.

Can author clarify their hypothesis besides giving a random observation without proper experiments “A direct comparison of the rate values is however difficult given the different microbial indicator used and, especially, the likely higher light attenuation experienced in filtrated algae pond water (potentially reducing the effect of direct DNA damage) and the potential presence of photosensitizers and radical scavengers in this medium (with unknown net effect on photo-oxidation).”

The similar for here as well please clarify this statement with the proper reference “The impact dissolved oxygen (DO) concentration on sunlight-mediated removal could not be investigated in laboratory assays due to difficulties in reaching HRAP supersaturated DO levels under aseptic conditions. The addition of known photosensitizers was not tested as this manipulation would artificially inflate exogenous photo-oxidation and, therefore, misrepresent the significance of this mechanism in real conditions”.

What was other conditions in experiment one and experiment 2 rather than mentioned in Table 2 please stated clearly in manuscript as DO concentration had no apparent impact on E. coli decay at neutral pH during Experiment 2, high DO concentrations were associated with improved E. coli decay under sunlight and darkness during Experiment 1.

6. PLOS authors have the option to publish the peer review history of their article (what does this mean?). If published, this will include your full peer review and any attached files.

Reviewer #1: **Yes: **vandana vinayak

Reviewer #2: No

---

## [Author Response · Author response to Decision Letter 0]

2 Mar 2022

GENERAL REPLY

We thank the reviewers for the valuable points they raised. Reviewer 1’s main criticism was that our study could have investigated the removal of other pathogen indicators. Reviewer 2 asked us to clarify several discussion points and cite additional literature.

All Reviewers’ comments are addressed individually below. The lines references provided correspond to the line numbering in the corrected version of the manuscript with tracked changes.

REPLY TO COMMENTS

REVIEWER #1:

Authors have presented a very interesting study on targeting a protocol for removal of pathogenic bacteria from wastewater used as a medium for growth of microalgae. I have few queries to which authors may reply:

1. Why only one bacteria was tested since in wastewater there are other pathogenic and non pathogenic bacteria present too.

We fully acknowledge that many pathogenic and non-pathogenic bacteria are present in wastewater and that the conclusions from our studies are only specific to the pathogen indicator Escherichia coli, as discussed in the manuscript L 548 – 554. Our scope was limited to human pathogens (the only relevant to wastewater discharge regulation) and due to resource limitations, we focused on a single indicator. E. coli was selected due to its relevance in water disinfection technology as indicated in our introduction (L 64 – 65).

We took great care to not generalize our conclusions and agree further investigations are needed to identify potential differences between E. coli and other indicators. To further lay emphasis on this point, the following sentence was added in the revised manuscript (L 552 – 554):

“Similar studies should therefore be carried out to investigate the removal of different indicators during HRAP based wastewater treatment.”

2. If the study showed that E coli decay at alkaline pH and dark decay mechanism which although authors have shown statistically , however, to anyone studying the algae related cultivation using wastewater understanding the physiology of this study is very important. Authors should add up what is the biology behind the adaption of this behavior.

We respectfully apologize for not being sure to fully understand the question. Extensive characterization of the microbial communities found in the wastewater effluent and HRAP broth was beyond the scope of our study focusing on the human pathogen indicator E. coli. We noted, L127 – 129, that “Prior monitoring showed Scenedesmus spp. was the dominant species in the HRAP (unpublished data) but this was not verified during bench assays”. We agree that ecology may impact the type and rates of dark removal mechanisms experienced, such as predation or toxicity from algae metabolites. We could not evidence any impact from the presence of algal metabolites during our laboratory study but noted this finding should not be broadly extrapolated (L214 – 220). We could not investigate predation for practical reasons, as noted in the manuscript in Table 1 and L 417 – 421, and later indicated this should be attempted in future studies (L 515 – 518). We hope these few precisions are answering Reviewer 1’s comment.

3. Further another important question is whether this study is useful for other algae also or only for Scenedesmus spp. Why many microalgae species which grow under alkaline conditions face bacterial infection, authors may throw some reasoning at this aspect so that the technology and protocol can be adapted by the readers for other algae too.

We again respectfully note our scope was the study of human pathogens rather than algae pathogens as the comment on “microalgae [facing] bacterial infection” may infer. Focusing on microalgae-based wastewater treatment, we believe our main finding on the significance of E. coli dark decay can be extrapolated to other microalgae ecologies. We however agree that the magnitude of day-time pH increase and episode of algal-metabolite toxicity are likely to be impacted by the type and activity of the microalgae found in the system. This is now more clearly expressed L 521 – 525.

“It is also critical to consider the findings from this study may not apply to HRAPs operated under significantly different conditions (e.g. different climates, pH control through CO2 addition, Mehrabadi et al. 2017) and/or hosting different microalgae ecology (e.g. influencing pH variations (57) potentially generating toxic metabolites (48)”. 

With regards to microalgae infection by bacteria, we note that the type of bacteria infecting microalgae grown at high pH should be physiologically very different than pathogen infecting the human body (pH 5-7). We therefore prefer to not speculate on this particular topic.

REVIEWER #2:

High-rate algae ponds (HRAPs) are wastewater treatment systems that enable combining cost-efficient secondary treatment at small scale with the production of a harvestable biomass for subsequent valorisation (e.g. biofuel). However, there is still limited data on pathogen removal during long-term HRAP operation with real effluents. This manuscript showed the potential significance of mechanisms driving pathogen removal in lab scale and bench scale in light of the specific environmental conditions occurring in HRAPs. However, Authors should improve this by following comments:

Starvation and heat inactivation:

You have good observation in this section but even at less temperature E coli including other bacteria can still survive and flourish in photobioreactors with monoculture of microalga. Please justify with more references.

We agree that although we never observed E. coli growth in the dark, our experimental observations do not exclude the possibility for this bacterium, or other pathogens, to grow under other conditions. We also acknowledge that E. coli regrowth has been reported at high temperature in tropical ponds. A key attribute of pathogen indicators used for wastewater management is precisely not to be able to grow under the conditions typically experienced during wastewater treatment although, again, conditions can become ‘atypical’ in certain climates in ponds. We added the following sentence to discuss this possibility L209 - 213:

“While E. coli re-growth has been reported in some systems, particularly in warm tropical ponds (26), no significant increase in E. coli cell count was noticed during this study, even at the warmest temperature tested (Fig S7-2 in S7 Appendix). Caution is therefore mandated before ruling out the possibility of re-growth, for E. coli or other indicators, depending on the conditions experienced.”

Algal-metabolite section is still unclear please addon this with few references.

The following sentence was added L214 – 216:

“despite previous evidence that several species of green microalgae can excrete a wide array of antibacterial compounds harmful to E. coli (27)”.

For clarity, the sentence directly following was amended as follows (L216 – 220):

“Algae-based toxicity was therefore unlikely significant under the conditions tested. This finding cannot be broadly extrapolated in view of the geographical and temporal variability in algal diversity found in HRAPs (28), and the diverse potential bactericidal compounds each species may be secreting (27,29)”.

Can author clarify their hypothesis besides giving a random observation without proper experiments “A direct comparison of the rate values is however difficult given the different microbial indicator used and, especially, the likely higher light attenuation experienced in filtrated algae pond water (potentially reducing the effect of direct DNA damage) and the potential presence of photosensitizers and radical scavengers in this medium (with unknown net effect on photo-oxidation).”

There is no hypothesis behind this statement. We first felt it was important to compare our rates with other values reported in the literature. As reported in the manuscript L 281 – 290, we found 2 studies reporting rates obtained in relatively similar experiments which were within the same order of magnitude as found during our study. We nevertheless felt this comparison was not sufficient to establish a trend and wanted to warm the readers that differences in experimental conditions and indicators (as identified for the two studies cited) could lead to very different results (meaning the ‘similarity’ we report in the rates may just be accidental). Admittedly, it is challenging to determine how incubating micro-organisms in a filtrate may increase or mitigate sunlight dependent decay. Consequently, we propose to remove the initial sentence cited below to avoid speculative statements (L290 – 293). As authors, we feel that this sentence should remain in the manuscript, but we leave it to the appreciation of Reviewer 2 and the Editor to decide if the sentence should be removed or not.

“the likely higher light attenuation experienced in filtrated algae pond water (potentially reducing the effect of direct DNA damage) and the potential presence of photosensitizers and radical scavengers in this medium (with unknown net effect on photo-oxidation)”

The similar for here as well please clarify this statement with the proper reference “The impact dissolved oxygen (DO) concentration on sunlight-mediated removal could not be investigated in laboratory assays due to difficulties in reaching HRAP supersaturated DO levels under aseptic conditions. The addition of known photosensitizers was not tested as this manipulation would artificially inflate exogenous photo-oxidation and, therefore, misrepresent the significance of this mechanism in real conditions”.

Our comment about dissolved oxygen (DO) only expresses the practical limitation that we could not carry out an experiment that would require, we believe, constantly bubbling our cultures with O2-enriched air to reach values as high as found in HRAPs (up to 300 % of the atmosphere saturation value, Sutherland et al. 2014). This complex experiment was deemed unnecessary based on our experimental data, so no reference is needed here.

With regards to the addition of photosensitizers, recording a positive impact (on decay) of adding known photosensitizer would simply confirm the known action of these chemicals (that photosensitizer generate radicals causing decay) but this confirmation would neither validate nor invalidate the possibility that unknown molecules found in the broth indeed acted as photosensitizers. In other words, we would likely record significant exogenous photo-oxidation in the presence of known added photosensitizer, but recording this mechanism in this artificial setting would not evidence that the mechanisms remains significant in the absence of added chemical.

The sentence L 322 – 330 was reworded as:

“The specific impact of dissolved oxygen (DO) concentration on sunlight-mediated removal could not be practically investigated in laboratory assays due to technical difficulties in reaching HRAP supersaturated DO levels (up to 300 % of the atmospheric saturation values (34)) under aseptic conditions. The addition of known photosensitizers was not tested as this manipulation would at best only verify that these known chemicals generate radicals enhancing E. coli removal, without evidencing that the mechanism is indeed taking place in the absence of the added chemicals.”

What was other conditions in experiment one and experiment 2 rather than mentioned in Table 2 please stated clearly in manuscript as DO concentration had no apparent impact on E. coli decay at neutral pH during Experiment 2, high DO concentrations were associated with improved E. coli decay under sunlight and darkness during Experiment 1.

We thank Reviewer 2 for highlighting that the data reported here may be confusing. We acknowledge there was significantly higher decay in Reactor A than Reactor B during the first bench experiment and that the only condition qualitatively different between the reactors during each experiment was DO concentration (similar temperature, high pH, and light conditions). Yet, we concluded DO had no impact because:

- DO had no impact during Experiment 2, when it was again the only discriminatory parameter between the 2 reactors (similar temperature, neutral pH, and light conditions).

- the literature consensus is that DO is not toxic by itself but only enhances ROS formation under sunlight. Yet, during Experiment 1, decay was also higher at high DO than low DO in the dark. This increased decay under darkness suggests the differences recorded may have been caused by another factor than DO. We therefore suggest the higher decay recorded in Reactor A during Experiment 1 was caused by the slightly higher temperature recorded in this reactor enhancing alkaline pH induced E. coli decay, as shown during our laboratory experiments. We propose the following changes in the paragraph “Impact of DO concentration” (L359 – 378) to clarify this: 

“While DO concentration had no apparent impact on E. coli decay at neutral pH (Experiment 2), high DO concentrations were associated with improved E. coli decay at high pH (Experiment 1). Critically, these differences (or lack of) were reported regardless of if the flasks were exposed to light or incubated in darkness, which disagrees with past literature showing DO concentration impacts E. coli via the formation of ROS (reactive oxygen species) under sunlight radiation but does not impact E. coli survival in the dark (31,44). Light-mediated ROS formation should have therefore enhanced higher E. coli decay only under light exposure and this effect should have been see in both Experiments 1 and 2 under sunlight, as opposed to the present observations showing not impact of light, but an interplay between the impacts of pH and DO concentration (Experiment 1 versus 2). The positive impact of DO concentration at high pH (Experiment 1) was instead likely due to the higher temperatures experienced in Reactor A than Reactor B. Indeed, Reactor A was slightly smaller than Reactor B, meaning higher temperatures were typically experienced in Reactor A (high DO) than in Reactor B (low DO). Since alkaline pH-induced toxicity to E. coli is sensitive to temperature (as evidenced during laboratory assays), temperature likely enhanced alkaline pH-induced toxicity more in Reactor A than in Reactor B during Experiment 1 conducted at high pH. This positive impact of temperature on alkaline pH-induced toxicity was logically not significant during Experiment 2 conducted at neutral pH. DO concentration is therefore concluded to have no significant impact on E. coli removal under the conditions studied.”

---

## [Editor Report · Decision Letter 1]

4 Mar 2022

Study from microcosms and mesocosms reveals *Escherichia coli* removal in high rate algae ponds during domestic wastewater treatment is primarily caused by dark decay

PONE-D-21-40922R1

Dear Dr. Paul,

We’re pleased to inform you that your manuscript has been judged scientifically suitable for publication and will be formally accepted for publication once it meets all outstanding technical requirements.

Kind regards,

Dr. Harish

Academic Editor

PLOS ONE
---

## [Editor Report · Acceptance letter]

9 Mar 2022

PONE-D-21-40922R1 

Study from microcosms and mesocosms reveals *Escherichia coli* removal in high rate algae ponds during domestic wastewater treatment is primarily caused by dark decay. 

Dear Dr. Chambonniere:

I'm pleased to inform you that your manuscript has been deemed suitable for publication in PLOS ONE. Congratulations! Your manuscript is now with our production department. 

Kind regards, 

on behalf of

Dr. Dr. Harish 

Academic Editor

PLOS ONE